# Exploring the Effect of Occurrence-Bias-Adjustment Assumptions on Hydrological Impact Modeling

**Jorn Van de Velde** [1,2,*], **Matthias Demuzere** [3], **Bernard De Baets** [2] **and Niko E. C. Verhoest** [1]

1   Hydro-Climatic Extremes Lab, Ghent University, 9000 Ghent, Belgium; niko.verhoest@ugent.be
2   KERMIT, Department of Data Analysis and Mathematical Modelling, Ghent University, 9000 Ghent, Belgium; bernard.debaets@ugent.be
3   Department of Geography, Ruhr-University Bochum, 44801 Bochum, Germany; matthias.demuzere@rub.de
*   Correspondence: jorn.vandevelde@ugent.be

**Abstract:** Bias adjustment of climate model simulations is a common step in the climate impact assessment modeling chain. For precipitation intensity, multiple bias-adjusting methods have been developed, but less so for precipitation occurrence. Intensity-bias-adjusting methods such as 'Quantile Delta Mapping' can adjust too many wet days, but not too many dry days. Some occurrence-bias-adjusting methods have been developed to resolve this by the addition of the ability to adjust too dry simulations. Earlier research has shown this to be important when adjusting on a continental scale, when both types of biases can occur. However, the newer occurrence-bias-adjusting methods have their weakness: they might retain a bias in the number of dry days when adjusting data in a region that only has too many wet days. Yet, if this bias is small enough, it is more practical and economical to apply the newer methods when data in the larger region are adjusted. In this study, we consider two recently introduced occurrence-bias-adjusting methods, Singularity Stochastic Removal and Triangular Distribution Adjustment, and compare them in a region with only wet-day biases. This bias adjustment is performed for precipitation intensity and precipitation occurrence, while the evaluation is performed on precipitation intensity, precipitation occurrence and discharge, which combines the former two variables. Despite theoretical weaknesses, we show that both Singularity Stochastic Removal and Triangular Distribution Adjustment perform well. Thus, the methods can be applied for both too wet and too dry simulations, although Triangular Distribution Adjustment may be preferred as it was designed with a broad application in mind.

**Keywords:** climate change impact; bias adjustment; occurrence-bias-adjustment; hydrological impact

## 1. Introduction

Climate change is one of the largest threats currently faced by society, with significant impacts on ecosystems, caused by the increase in naturally occurring hazards, such as droughts, wildfires, hurricanes and floods [1]. To assess how this increase in hazards is influenced by climate change conditions, a modeling chain consisting of Global Circulation Models (GCMs), Regional Climate Models (RCMs) and local impact models is commonly used [2]. The GCMs allow for the assessment of future climate conditions globally [3]. However, their scale is too coarse to be used directly in local impact models. This is especially true in hydrology, where local meteorological variables such as precipitation can differ substantially within a watershed. Therefore, RCMs are used to downscale the coarse-scale data to the local scale. This downscaling is conducted by physically simulating the local climate, using the GCM's output as boundary conditions. A prime example of the downscaling process is the CORDEX project [4], where RCMs with a grid resolution of 12.5 km are commonly used. Although the resolution of climate models is quickly becoming finer, the most recent models are not yet fit for long-term projections [5]. In the more commonly used models for impact assessment, such as the CORDEX RCMs, systematic errors still occur due to imperfect parameterizations, discretization and spatial

averaging within the grid boxes [6]. These errors lead to the presence of biases, most often in precipitation [7]. Biases are generally described as "a systematic difference between a simulated climate statistic and the corresponding real-world climate statistic" [8] and increase the uncertainty in the last step of the modeling chain, i.e., the local impact modeling, thus entailing the necessity to adjust them [8–10].

During the last 15 years, many methods have been developed to overcome the bias problem (see Teutschbein and Seibert [6], Gutiérrez et al. [11] for recent overviews). These methods, which adjust the mean, variance and/or the full distribution of the variable under consideration, are well-studied for hydrological impact studies (e.g., Addor and Seibert [12], Räty et al. [13], Pulido-Velazquez et al. [14]) and use a 'transfer function' to transfer the information from the observations to the simulations. Most of the common bias-adjusting methods belong to the family of quantile mapping methods [15]. Teutschbein and Seibert [6] found this family of methods to be the best-performing in comparison with other families and methods such as linear scaling' [16], 'local intensity scaling' [17] or power transformation [18]. This has become especially clear when comparing the influence of these different families on hydrological time series simulation [19,20]. Quantile mapping adjusts the full distribution of the variable's future simulations on the basis of the cumulative distribution functions (CDFs) of the historical simulations and the historical observations. A transfer function is composed from the CDFs, resulting in an adjusted future simulation for a value $x$ at time step $i$ in the time series:

$$x_i^{\text{fa}} = F_{X^{\text{ho}}}^{-1}\left(F_{X^{\text{hs}}}\left(x_i^{\text{fs}}\right)\right), \tag{1}$$

with ho, hs, fs and fa, respectively, the index of the historical observations, uncorrected historical simulations, uncorrected future simulations and adjusted future simulations, and $F_X$ a CDF.

Various extensions and variants of the standard quantile mapping method have been proposed. A recent variant is 'Quantile Delta Mapping' (QDM) [21–23]. It was introduced to better retain the trends of the climate model and though being criticized [24], it is increasingly used, especially as a step in more complex multivariate bias-adjusting methods [25–27]. Other quantile mapping methods are also frequently used, such as 'CDF-transform' (CDF-t), which was originally proposed by Michelangeli et al. [28] and later used by e.g., Vrac [29], and standard empirical quantile mapping, used by e.g., Räty et al. [13] and Zscheischler et al. [30].

One overlooked aspect in many of the bias-adjusting methods is the adjustment of precipitation occurrence. Climate models often simulate too many rainy days, the so-called drizzle effect [31,32]. This has since long been acknowledged to be a problem, especially when the intermittence of rainfall is important [33], as is the case for hydrological impact assessment. As an example, consider that a rainy day may cause an increase in soil moisture. Subsequently, the infiltration capacity decreases, leading to more runoff and river flow, and possibly leading to flooding. Though most bias-adjusting methods implement some basic form of occurrence adjustment before adjusting the bias in intensity, there is limited research on the effect of the inclusion of occurrence-bias-adjustment. One of the earliest studies on the performance of bias adjustment with respect to transition probabilities was published by Rajczak et al. [34]. They found that quantile mapping corrects the frequency of wet days and that it also improves the transition probabilities and both dry and wet spell length. This implies that QDM will also perform well for the adjustment of wet days; as we will discuss, its design enables it to adjust the number of dry days. However, as Themeßl et al. [35] indicate, there are also many locations at which the model has more dry days than the observations, which QDM cannot simply correct. Thus, when in need of climate data on a continental scale, the too dry locations will not be adjusted when applying QDM. To enable a more flexible occurrence-bias-adjustment, Vrac et al. [36] introduced the 'Singularity Stochastic Removal' method (SSR). This method is able to adjust both too dry and too wet simulations and has been shown, in combination with CDF-t, to outperform other methods such as thresholding [17], the 'direct' method (application of

CDF-t without occurrence-bias-adjustment) and positive adjustment (which only adjusts precipitation intensities). In theory, SSR should also work fine with quantile mapping methods other than CDF-t. Yet, as it was specifically designed in combination with this method, it is unclear whether it works well in combination with QDM. In that case, it might actually introduce too few or too many dry days. As an alternative, we consider the recently introduced occurrence-bias-adjustment method called 'Triangular Distribution Adjustment' (TDA) [37]. This method has an element of stochasticity and can also correct both too wet and too dry time series. However, it has the same disadvantage as SSR, i.e., it might introduce too many dry days. Yet, if the remaining bias is small enough, TDA might be a powerful method for adjusting the occurrence over many grid cells with varying biases. Themeßl et al. [35] and Vrac et al. [36] already discussed the potential and importance of the capacity to adjust occurrence biases in areas where the models result in too dry series. Yet, it is unclear if the weakness has a large impact in the more generic situation of too wet models, and when studying local impact such as floods. When assessing floods, it is important to know whether the assumptions of the methods influence the hydrological impact and how large the impact is. If there is an impact, it might be more feasible to adapt the choice of bias-adjusting method to the occurrence bias of each grid box or region consisting of similar grid boxes when adjusting at a continental scale. If the impact is small, then it is far more economical to use the same method for all grid boxes, thus ensuring easier bias adjustment. Besides, when other methods than QDM are used, such as multivariate bias-adjusting methods [38], the occurrence-bias-adjusting methods ensure a thorough occurrence adjustment.

The goal of this study is thus to compare three bias-adjusting methods: QDM, SSR and TDA. Although SSR has been introduced a few years ago and was assessed on a continental scale, it is unclear whether SSR and similar methods are outperformed by the simpler QDM in the specific context of too wet simulations. If this is not the case, or QDM only slightly outperforms the other methods, then this provides an additional argument to apply them in a broad range of regions, regardless of the bias (too wet or too dry).

## 2. Data and Methods

### 2.1. Data

#### 2.1.1. Observations

The comparison in the present paper is performed by calibration on historical time series and validation on recent past time series. In order to perform a robust calibration and validation, the time series used have to be long enough [39]. Observational data were obtained from the Uccle observatory maintained by the Belgian Royal Meteorological Institute (RMI). The main time series used in the present paper is the 10-min precipitation amount, gauged with a Hellmann-Fuess pluviograph from 1898 to 2018. An earlier version of this precipitation dataset was described by Demarée [40] and analyzed by De Jongh et al. [41]. Multiple other studies have used this time series [41–45]. For the hydrological modeling, the precipitation time series was combined with a potential evaporation time series. The daily potential evaporation was calculated by the RMI from 1901 to 2019, using the Penman formula for a grass reference surface [46] with variables measured at the Uccle observatory. The 10-min precipitation time series was aggregated to daily level to have the same resolution as the evaporation time series. In total, 117 years of data were combined, from 1901 up to and including 2017.

#### 2.1.2. Climate Simulations

For the simulations, a set-up similar to Van de Velde et al. [47] was used. The Rossby Centre regional climate model RCA4, part of the EURO-CORDEX project [4], was used [48] as it is one of the few RCMs with potential evaporation as an output variable. This RCM is forced with boundary conditions from the MPI-ESM-LR GCM [49]. Historical data and scenario data for the grid cell comprising Uccle were respectively obtained for 1970–2005 and 2006–2100. The former time frame is limited by the earliest available data from the

RCM. The latter time frame was only used until 2017, in accordance with the observational data. As climate change scenario, an RCP4.5 forcing was used in the present paper [50]. This forcing does not have a large impact, since only 'near future' (from the model point of view) data were used. As a single RCM-GCM model chain was used in the present paper, it is impossible to discuss the uncertainties related to climate modelling. Nonetheless, this allows us to clearly focus on the uncertainties introduced by the assumptions of the bias-adjusting methods.

### 2.2. Bias-Adjusting Methods

### 2.2.1. Quantile Delta Mapping

Quantile Delta Mapping (QDM) was first proposed by Li et al. [21] (as 'Equidistant CDF-matching') and was extended by Wang and Chen [22] (as 'Equiratio CDF-matching') to better handle precipitation adjustment. Cannon et al. [23] combined these methods, resulting in 'Quantile Delta Mapping'. Mathematically, QDM can be written as

$$x_i^{\text{fa}} = x_i^{\text{fs}} \frac{F_{X^{\text{ho}}}^{-1}\left(F_{X^{\text{fs}}}\left(x_i^{\text{fs}}\right)\right)}{F_{X^{\text{hs}}}^{-1}\left(F_{X^{\text{fs}}}\left(x_i^{\text{fs}}\right)\right)} \tag{2}$$

for a bounded variable such as precipitation. For evaporation, the few available studies (e.g., Lenderink et al. [16]) suggest the same adjustment as for precipitation.

To ensure the consistency of the time series, a 91-day moving window is chosen in this study, as suggested by Rajczak et al. [34] and Reiter et al. [51]. This enables the adjustment of each day based on 91 days/year · 20 years = 1820 days. These days were used to build an empirical CDF (as in Gudmundsson et al. [20], Gutjahr and Heinemann [52], among others), because of the ease of application. It is also important to note that for precipitation, Equation (2) was applied only on the days considered wet, i.e., with a precipitation higher than 0.1 mm. For consistency, a threshold of 0.1 mm was also used for evaporation. However, the wet days can still become dry (e.g., with precipitation amount < 0.1 mm) if the ratio in Equation (2) is small enough. This way, QDM will always adjust the number of dry days in the model to that of the historical observations. This adjustment can be clarified by Figure 1, which is based on the data used in the present paper: for each of the $n$ observed dry days (e.g., $F_{X^{\text{ho}}}\left(x^{\text{ho}}\right) < 0.35$), the value $F_{X^{\text{ho}}}^{-1}\left(F_{X^{\text{fs}}}\left(x_i^{\text{fs}}\right)\right)$ will be equal to zero. As this is the numerator in Equation (2), the $n$ lowest simulated days, with precipitation depths ranging from 0 to 0.8 mm, will also become dry.

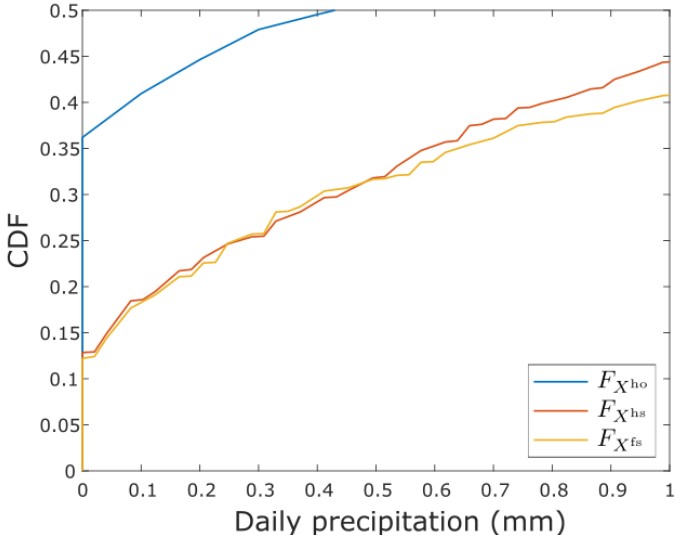

**Figure 1.** CDFs of the historical observations, historical simulations and future simulations used in this study for daily precipitation depths ranging between 0 and 1 mm.

The weakness of QDM is that the described adjustment can only be applied if the observations have more dry days than the model simulations. If the opposite occurs, the denominator of Equation (2) will be zero for all historically simulated dry days, and thus no adjusted value can be calculated. To deal with this problem, other methods have to be used in a preprocessing step, such as SSR or TDA.

### 2.2.2. Singularity Stochastic Removal

Although Themeßl et al. [35] first discussed the idea for an adjustment that could handle an excess of dry days in the model simulations, the 'Singularity Stochastic Removal' (SSR) [36] is the first method to flexibly do so, incorporating both the adjustment of too wet and too dry model simulations. This method temporarily removes the zeroes from all time series and reintroduces the zeroes after the bias adjustment is applied, based on the frequency method mentioned by Cannon et al. [23] and a method used to alter temperature data by Zhang et al. [53].

The zeroes are removed by calculating the lowest strictly positive amount of rain $P_{min}$ in any of the time series used, including the historical observations, historical simulations and future simulations. For all days with rainfall amounts below $P_{min}$, a new rainfall depth $P_{new}$ is randomly drawn from a uniform distribution on $]0, P_{min}[$. There are thus no longer dry days, or singularities, in the time series, hence the method's name. The singularities are now transformed into unique values, thereby enabling an easy transformation: problems due to the denominator becoming zero cannot occur. For the calculation of $P_{min}$, we have applied SSR on a monthly basis. This is a departure from Vrac et al. [36], where all data were adjusted simultaneously for the time series, but enables a better comparison with TDA, which is also applied monthly. Over all repetitions and months, $P_{min}$ ranged from 0.0103 to 0.0412 mm. After the application of the intensity-bias adjustment, a post-processing step is applied: all days of $X^{fs}$ with a rainfall amount below $P_{min}$ are set to zero. The full procedure is summarized in Algorithm 1.

There is one important issue with SSR: it was designed for CDF-t. Although Vrac et al. [36] state that the method can be combined with quantile mapping methods other than CDF-t, this might be ill-suited in practice. CDF-t adjusts the CDF according to the following formula: $F_{X^{fa}} = F_{X^{ho}}\left(F_{X^{hs}}^{-1}\left(F_{X^{fs}}\left(x^{fs}\right)\right)\right)$. In contrast, QDM uses the relative difference between two CDFs (as displayed in Equation (2)), a difference that is changed by SSR's introduction of small values. This implies that some currently wet days will be transformed into dry days, or that too few wet days will be transformed. However, the number of days cannot be exactly determined and depends on the CDFs of the historical observations, historical simulations and future simulations. As such, it is much harder to exactly control the adjustment by SSR when it is combined with QDM. However, it is unclear how large this impact will be.

### 2.2.3. Triangular Distribution Adjustment

Triangular Distribution Adjustment (TDA) is a method that was developed in an attempt to easily and simultaneously add and remove dry days while being stochastic [37]. In contrast to SSR, it is not designed with a specific method in mind, but rather as a stand-alone occurrence-bias-adjusting method. The main idea of TDA is to introduce both stochasticity and a threshold to preserve precipitation by using a distribution.

As a first step, the number of days to be removed from or added to the future time series is calculated using the ratio of the historical observed and simulated dry day frequency, respectively $f^{ho}$ and $f^{hs}$. This ratio is assumed to be the same for the future scenarios so that it can be used to calculate a corrected future dry day frequency, $f^{fa}$, which is in turn used to calculate the number of days $\Delta N$ to be removed or added. A shortcoming of this method is that $f^{fa}$ might become higher than 1 or lower than 0 if the bias between $f^{ho}$ and $f^{hs}$ were too large. In such cases, $f^{fa}$ should be bounded to either 1 or 0, implying that the future month under consideration has either no or only wet days.

---

**Algorithm 1** Singularity Stochastic Removal.

---

**Input:**

   Historical observations $X^{\text{ho}}$

   Historical simulations $X^{\text{hs}}$

   Future simulations $X^{\text{fs}}$

**Output:**

   Adjusted future simulations $X^{\text{fs}}_{\text{out}}$

   {Before the intensity-bias adjustment}

   **for** $m = 1 : 12$ **do**

      Select the data for month $m$: $X^{\text{ho}}_m$, $X^{\text{hs}}_m$ and $X^{\text{fs}}_m$

      Determine the length of the monthly time series $N_{\text{days}}$

      Determine $P_{\text{min}}$ based on $X^{\text{ho}}_m$, $X^{\text{hs}}_m$ and $X^{\text{fs}}_m$

      **for** $i = 1 : N_{\text{days}}$ **do**

        **if** $x^{\text{ho}}_i < P_{\text{min}}$ **then**

          Simulate a new value

        **end if**

        **if** $x^{\text{hs}}_i < P_{\text{min}}$ **then**

          Simulate a new value

        **end if**

        **if** $x^{\text{fs}}_i < P_{\text{min}}$ **then**

          Simulate a new value

        **end if**

      **end for**

   **end for**

   {After the intensity-bias adjustment}

   Set all values $x^{\text{fs}} < P_{\text{min}}$ to 0, yielding $X^{\text{fs}}_{\text{out}}$

---

The next step is to remove or add the required number of dry days. The triangular distribution is used to ensure that the removal or addition of dry days does not change the extremes, hence the method's name. Other distributions can also be implemented, but the triangular distribution is chosen for its ease of implementation. The general idea of this method is given in Figure 2. The CDF of the triangular distribution is given by

$$T(x) = \begin{cases} 0 & \text{if } x < 0 \\ 1 - \frac{(b-x)^2}{b^2} & \text{if } 0 \leq x < b \\ 1 & \text{if } b \leq x \end{cases} \tag{3}$$

The single parameter of the triangular distribution, $b$, is the threshold that determines precipitation conservation. It corresponds with a precipitation amount $x_{\text{thr}}$, which can be calculated as follows:

$$x_{\text{thr}} = F^{-1}_{X^{\text{fs}}}(b \mid x > 0.1), \tag{4}$$

with $F_{X^{\text{fs}}}(x \mid x > 0.1)$ the CDF of the precipitation of wet days. In this study, $b$ was set to 0.9, which ensures that the highest extremes are never changed to dry days, while not completely restricting the choice of days. This implies that relatively wet days can also become dry, thereby adjusting the temporal structure of the precipitation time series. This adjustment can be a strength, if the originally simulated time series has a poor temporal structure. Although the threshold is quite high in the present paper, the probability that days becomes dry decreases rapidly with increasing precipitation amounts.

Dry days are added as follows, with steps 1–2 applied for the number of dry days $\Delta N$:

1. Choose a day $t$ (with precipitation $x_t > 0.1$) randomly from the wet day time series. This day has a corresponding cumulative probability of $\xi = F_{X^{fs}}(x_t \mid x > 0.1)$.
2. Sample $k$ from a uniform distribution on $[0,1]$. If $T(\xi) < k$, then draw $x_t$, the new value for day $t$, from a uniform distribution on $[0, 0.1]$. $x_t$ is drawn randomly, to take into account that model simulations hardly have zero values. If $T(\xi) > k$, then repeat from step 1.

If dry days need to be removed, the value of $\xi$ is used, which is calculated using Equation (3):

$$\xi = b\left(1 - \sqrt{1 - T(\xi)}\right). \tag{5}$$

In this case, the only restriction on the choice of days is that they have to be dry. Thus $\Delta N$ dry days will be randomly selected from the time series and removed, without considering the temporal structure of the time series. The process of dry day removal is given as follows:

1. For every dry day to be removed, choose a dry day $t$ randomly from the dry day time series (with $x_t < 0.1$) and sample $k$ from a uniform distribution on $[0, 1]$.
2. Calculate $\xi = b\left(1 - \sqrt{1 - k}\right)$.
3. Set $x_t = F_{X^{fs}}^{-1}(\xi) = F_{X^{fs}}^{-1}\left(b\left(1 - \sqrt{1 - k}\right) \mid x > 0.1\right)$.

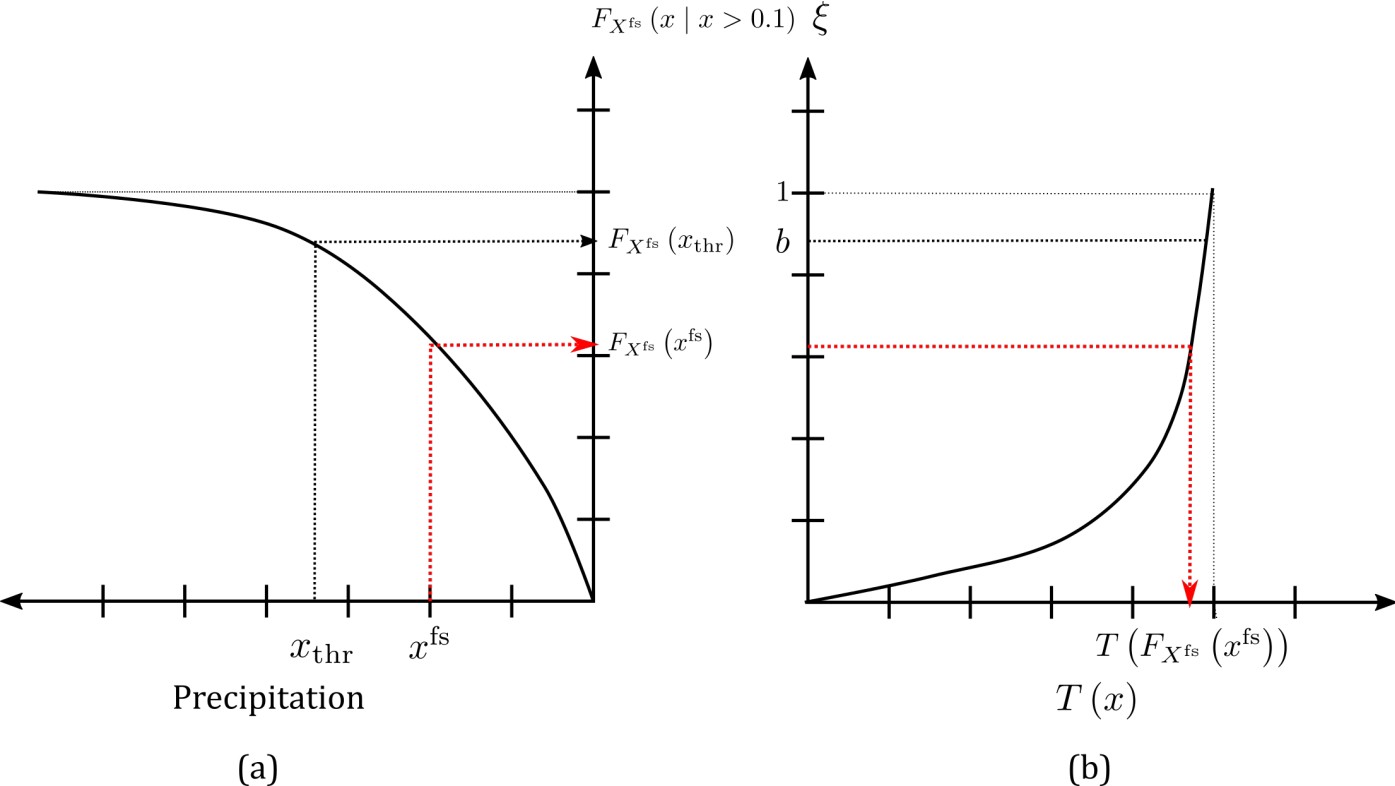

**Figure 2.** Overview of the distributions used in TDA: (**a**) CDF of the precipitation of the wet days in the future simulation, (**b**) CDF of the triangular distribution. The red arrow displays how a value $x^{fs}$ is transformed into a value $T(\xi)$ in the wet-to-dry case. The black arrow indicates the threshold value. Adapted from Pham [37].

Similar to SSR, TDA is applied on a monthly basis and for both historical and future simulations, which results in Algorithm 2.

---

**Algorithm 2** Triangular Distribution Adjustment.

---

**Input:**
  Historical observations $X^{\text{ho}}$
  Historical simulations $X^{\text{hs}}$
  Future simulations $X^{\text{fs}}$
**Output:**
  Adjusted future simulations $X^{\text{fs}}_{\text{out}}$

  Initialization
  **for** $m = 1 : 12$ **do**
    Select data for month $m$: $X^{\text{ho}}_m$, $X^{\text{hs}}_m$ and $X^{\text{fs}}_m$ {Loop over months}
     {Dry day frequency calculation}
    Calculate $f^{\text{hs}}$, $f^{\text{ho}}$ and $f^{\text{fs}}$
    Calculate $f^{\text{fa}} = f^{\text{fs}} \cdot f^{\text{ho}} / f^{\text{hs}}$
    Calculate the difference in dry days $\Delta N$
     {Empirical CDF}
    Select the wet days for month $m$
    Select the dry days for month $m$
    Calculate the empirical CDF $F_{X^{\text{fs}}}$ for the wet days
     {Adjustment}
     {Addition of dry days}
    **if** $\Delta N > 0$ **then**
      Set counter to 1
      **while** counter $\leq \Delta N$ **do**
        Randomly select a day $t$
        Calculate $\xi$ using $x_t$
        Randomly select a value k
        **if** $T(\xi) < k$ **then**
          Randomly replace the value of day $t$ with a value on [0,0.1]
          Add 1 to the counter
        **end if**
      **end while**
       {Removal of dry days}
    **else if** $\Delta N < 0$ **then**
      Set counter to 1
      **while** counter $\leq | \Delta N |$ **do**
        Randomly select a day $t$
        Calculate $\xi$
        Calculate $x_t$
        Add 1 to the counter
      **end while**
    **end if**
    Recombine the wet and dry day time series in the original order for month $m$
    Reintroduce the adjusted data for month $m$ in the original time series
  **end for**

---

The advantages introduced by this method come along with a weakness: it may introduce too many dry days in combination with quantile mapping. As the method randomly selects days from the time series, some days with a small precipitation amount will remain. Although TDA perfectly adjusts the number of dry days, extra dry days can be introduced because of the bias adjustment of these remaining days with low precipitation.

QDM will map them according to the CDF of the historical observations. The reason for this lies, as can be seen in Figure 1, in the fact that this method interpolates the CDF between 0 and 0.1 mm, where no real observations occur. As the remaining low precipitation days are mapped to these interpolated values, QDM creates extra days with a precipitation amount <0.1 mm, and hence, extra dry days are introduced as these values are set to zero. The number of extra days introduced is hard to predict and depends on the selection of days by TDA. The question is therefore whether or not this weakness outweighs the flexibility.

*2.3. Evaluation Strategy*

To compare and evaluate different bias-adjusting methods, a logical evaluation structure is required. To offer detailed information on the performance of the bias-adjusting methods in a realistic set-up, they were applied in a present-day climate change context. For this application, 1970–1989 was chosen as the control or 'historical' time period and 1998–2017 as the validation or 'future' period. Choosing these time frames allowed for a comparison with observations in the validation period for a time period that was already affected by climate change [54]. For a robust calculation of the bias adjustment, 30 years of data are advised [39,51]. This decreases the effect of internal variability of the climate model and therefore decreases the uncertainty of the bias adjustment results [55]. However, 30 years of data would in this specific case have resulted in overlapping time series, as no earlier data are available. This would have consequently decreased the differences between the time series. The 20 years chosen as an alternative were used in a set-up called a 'pseudo-projection' (e.g., Li et al. [21]). This evaluation set-up resembles the 'Differential Split-Sample Testing' (DSST) implemented by Teutschbein and Seibert [56], which is based on the work of Klemeš [57]. In DSST, the calibration and validation time series are chosen to make the difference between both as large as possible, which allows to study how robust a model is to changes.

Besides the choice of calibration and validation years, it is important to define the indices used for the evaluation (Table 1). As discussed by Maraun et al. [58] and Maraun and Widmann [59], it is important to use indices that are both directly and indirectly affected by the bias adjustment. They argue that only using directly adjusted indices can possibly be misleading, as those indices are used to build the transfer function in the calibration period. Thus, applying the transfer function in the projection period should also adjust these indices. To account for this, indices based on the discharge $Q$ were used ($Q_x$ and $Q_{T20}$). These indices allow for the assessment of the impact of the occurrence-bias-adjusting methods on simulated river flow. As a final important group of indices, occurrence indices were used, in order to assess how the combined methods differ in changing the precipitation occurrence of the time series (indices $P_{P00}$, $P_{P10}$, $N_{dry}$ and $P_{lag1}$). For the precipitation amount, the percentiles of the empirical distributions were considered. Before calculating the indices, a basic thresholding was applied. All days with a precipitation depth < 0.1 mm in the simulated time series were set to a precipitation depth of 0 mm.

**Table 1.** Overview of the indices used.

| Index | Name |
|---|---|
| $P_x$ | Precipitation amount percentiles, with $x$ the percentile considered |
| $Q_x$ | Discharge percentiles, with $x$ the percentile considered |
| $Q_{T20}$ | 20-year return period value of discharge |
| $P_{P00}$ | Precipitation transition probability from a dry to a dry day |
| $P_{P10}$ | Precipitation transition probability from a wet to a dry day |
| $N_{dry}$ | Number of dry days |
| $P_{lag1}$ | Precipitation lag-1 autocorrelation |

Similar to Pham et al. [60] and Van de Velde et al. [47], we used the 'Probability Distributed Model' (PDM, [61]), a lumped conceptual rainfall-runoff model to calculate the discharge for the Grote Nete watershed in Belgium. This model uses precipitation

and evaporation time series as inputs to generate a discharge time series. The PDM as used here was calibrated (RMSE = 0.9 m$^3$/h, see Pham et al. [60] for more details) using the Particle Swarm Optimization algorithm (PSO, [62]). As in Pham et al. [60] and Van de Velde et al. [47], it was assumed that the differences between meteorological conditions in the Grote Nete-watershed and Uccle were negligible, allowing for the use of the adjusted data for the Uccle grid cell as forcing for the PDM. To calculate the bias on the indices, observed, raw and adjusted RCM time series were used as forcing for this model. The discharge time series generated by the observations is considered to be the 'observed' discharge, and biases are calculated in comparison with this time series.

To summarize and compare the performance of the methods for each of the index groups, the residual biases for the indices of QDM, SSR & QDM and TDA & QDM were calculated, based on the 'added value' concept (discussed in Di Luca et al. [63]). The residual bias was introduced by Van de Velde et al. [47] and can be calculated relative to the model bias or the observations. This enables a detailed comparison based on how well the methods perform at removing the bias and the size of the bias removal in comparison with the original value for the corresponding index for the observation time series. The residual bias relative to the observations RB$_O$ for an index $k$ is calculated as follows:

$$\mathrm{RB}_{Ok} = 1 - \frac{|\,\mathrm{bias}_{\mathrm{raw},k}\,| - |\,\mathrm{bias}_{\mathrm{adj},k}\,|}{|\,\mathrm{obs}_k\,|}, \tag{6}$$

with obs$_k$, bias$_{\mathrm{raw},k}$ and bias$_{\mathrm{adj},k}$ respectively the value of the observations for index $k$, the bias of the raw climate model simulations and of the adjusted climate simulations for index $k$. The residual bias relative to the model bias RB$_{MB}$ for an index $k$ is calculated as follows:

$$\mathrm{RB}_{MBk} = 1 - \frac{|\,\mathrm{bias}_{\mathrm{raw},k}\,| - |\,\mathrm{bias}_{\mathrm{adj},k}\,|}{|\,\mathrm{bias}_{\mathrm{raw},k}\,|}. \tag{7}$$

If the values of the residual biases are lower than 1, the method performs better than the raw RCM. For RB$_O$, it is possible to have values lower than 0 in case the value after bias adjustment is exceptionally small in comparison to the value of the observations. The best methods have low scores on both residual biases for their indices.

The original data used for calculating the residual biases, i.e., the observational data, the raw climate simulations and the effective biases of the adjusted simulations, can be found in Appendix A.

### 2.4. Calculation Set-Up

Both SSR and TDA were applied as a preprocessing step before the application of QDM, but in case of SSR an additional postprocessing step was also included. As the occurrence-bias-adjusting methods include stochasticity, the calculations were repeated 20 times to account for variability. After the repetitions were carried out, the value of each index was first calculated on the basis of every simulation. Then, the values were averaged over the simulations; these averaged values were used for the comparison of the methods. Biases on the indices are always calculated as simulations minus observations, indicating a positive bias if the simulations are larger than the observations and vice versa.

### 3. Results

### 3.1. Precipitation Intensity

Figure 3 shows that the RB$_O$ and RB$_{MB}$ values are lower than 1 for most indices. This was to be expected: QDM has already been proven to be a successful intensity-bias-adjusting method. However, the full effect of the occurrence-bias-adjusting methods on the residual biases cannot be discussed based on the plot. By construction, these methods adjust the lowest percentiles, which could not be plotted. In case of the 5th and 25th percentiles, the observed value and the climate model bias were 0, and in case of the 50th the remaining bias was too small, leading to a negative RB$_O$ value. This indicates that the

values are very similar. Only TDA & QDM performs slightly worse for the 25th percentile, adding a bias to the simulations, and slightly better for the 50th. Yet, this amounts to an absolute difference of respectively 0.02 and 0.04 mm in comparison with QDM and SSR & QDM. As such, both SSR & QDM and TDA & QDM perform similarly to QDM and the impact on precipitation intensity is practically non-existent.

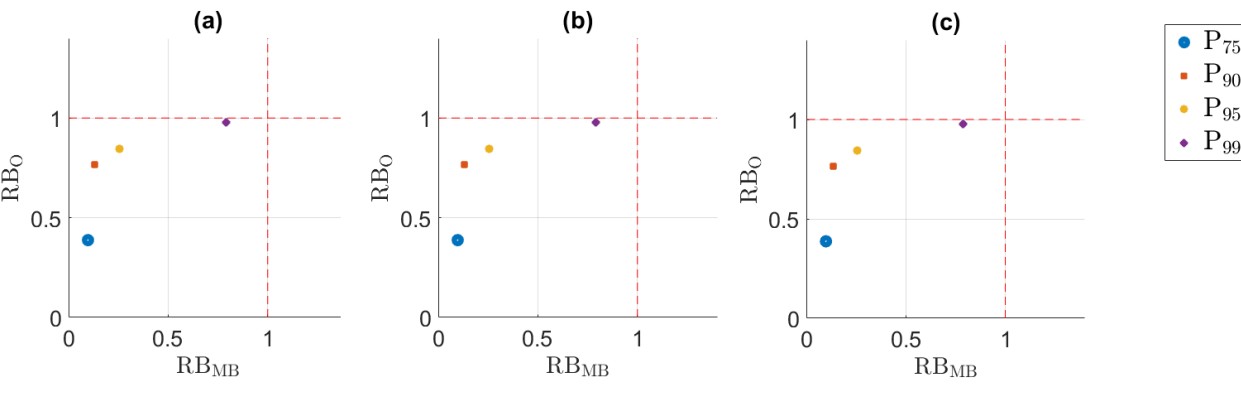

**Figure 3.** $RB_{MB}$ versus $RB_O$ for the precipitation intensity percentiles. (**a**) QDM, (**b**) SSR & QDM, (**c**) TDA & QDM.

Besides the impact of the occurrence-bias-adjusting methods, Figure 3 also shows that QDM performs worse for the higher quantiles. For the 99th percentile, the biases after adjustment are even worse than for the raw climate model, with an absolute bias of more than 4 mm. This could be caused by bias-nonstationarity due to climate change, which is discussed in depth by Van de Velde et al. [47].

### 3.2. Precipitation Occurrence

For precipitation occurrence, the $RB_O$ and $RB_{MB}$ values are slightly influenced by the methods, as can be seen in Figure 4. SSR performs about as good as QDM: both the number of dry days and the dry-to-dry transition probability slightly differ, but in practice only the number of dry days is really influenced by the method. Yet, SSR only has a negative bias of 17 days (averaged over the 20 repetitions): the theoretical weakness of this method is thus not as pronounced as expected. Between QDM on the one hand and TDA & QDM on the other hand, the differences are larger for the dry-to-dry transition probability, the wet-to-dry transition probability, and the number of dry days. However, in terms of absolute biases these differences are relatively small: TDA & QDM has a bias of 2% in both dry-to-dry and wet-to-dry transition probabilities and of 42.2 days (averaged over the 20 repetitions) for the number of dry days. As such, it appears that the theoretical differences between the methods do not have a large impact for the time series and location studied.

Based on Figure 4, it appears that the adjustment of the lag-1 autocorrelation is worse than the adjustment of the other indices. However, as the $RB_O$ and $RB_{MB}$ values are respectively below 1 and below 0.5, the performance is still relatively good. The residual bias is 2% for QDM and SSR & QDM and 3% for TDA & QDM. In comparison with the original bias of 11%, the residual bias is still relatively large, hence the higher $RB_{MB}$ value. Similarly, as the observed lag-1 autocorrelation is 33%, the $RB_O$ value remains large. Nevertheless, in comparison with other relative biases, the adjustment could have been better. This implies that although the general wet-dry structure of the time series is adequately simulated, the internal structure of the rain periods is not, and that the bias-adjusting methods cannot account for this. Of course, this result is case-specific: for other locations or combinations of simulations and observations, it might be that both the dry-wet structure and wet day structure are hard to adjust.

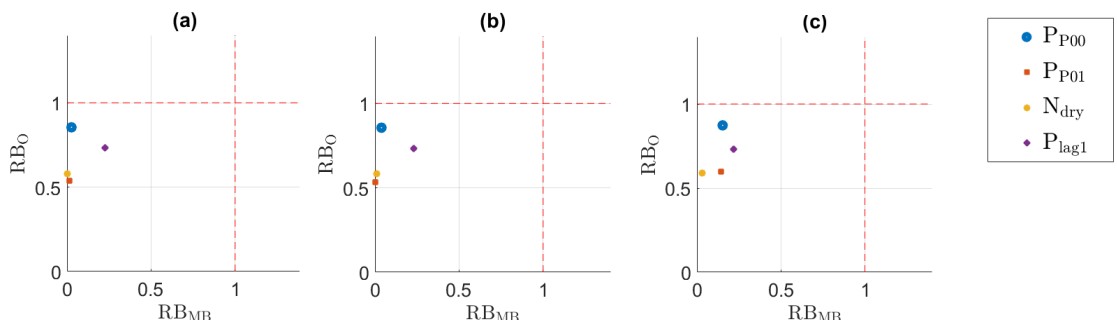

**Figure 4.** $RB_{MB}$ versus $RB_O$ for the precipitation occurrence indices. (**a**) QDM, (**b**) SSR & QDM, (**c**) TDA & QDM.

Not only for the lag-1 autocorrelation, but for all indices there is a difference in $RB_O$ and $RB_{MB}$ values. For $RB_O$, they are all above 0.5, while for $RB_{MB}$ they are all below 0.5, a difference caused by the discrepancies between the observation and the biases. The size of the bias removed is relatively large in comparison with the climate model bias, but small in comparison with the observations. This is made clear by $N_{dry}$, which has the best $RB_O$ value for all combinations: in comparison with the observed 3470 dry days, the (almost) removed bias of 1466 days has a large influence.

When comparing Figures 3 and 4, it can be seen that the $RB_O$ and $RB_{MB}$ values for precipitation occurrence cluster more in each plot. This may indicate that precipitation occurrence is more robust to climate change than intensity. Especially for TDA this matters, as the method depends on the stationarity of the ratio of historical and simulated dry day frequency. This is studied more in-depth by Van de Velde et al. [47], where it is indeed concluded that precipitation occurrence is more robust than precipitation intensity, at least for the time period under consideration.

### 3.3. Discharge

For discharge, the differences between the $RB_O$ and $RB_{MB}$ values of QDM, SSR & QDM and TDA & QDM are small, but non-negligible. For the highest discharge quantiles, Figure 5 shows that TDA performs slightly better. Although the difference between the methods is smaller than 1 m$^3$/s, it is not negligible: the discharge at the 99th percentile using observed values is 18.71 m$^3$/s. This could imply that it is better to use TDA than SSR for flood impact assessment. However, for the 20-year return period, the bias of TDA & QDM is larger than the bias of SSR & QDM: 8.42 m$^3$/s versus 7.71 m$^3$/s. Yet, as indicated by the figure, this difference is very small compared to both the original bias and the observed value. Besides, the values are similar to those of QDM. Thus, considering all percentiles, both SSR & QDM and TDA & QDM perform as good as QDM. The theoretical weaknesses and resulting small differences in precipitation occurrence have almost no influence on the discharge and do not allow for a clear decision on the method.

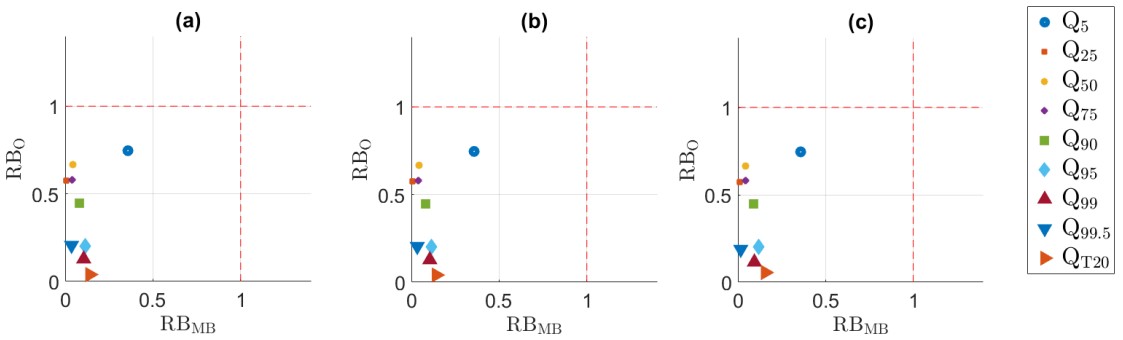

**Figure 5.** $RB_{MB}$ versus $RB_O$ for the discharge percentiles and the 20 year return period value. (**a**) QDM, (**b**) SSR & QDM, (**c**) TDA & QDM.

## 4. Discussion and Conclusions

In the present paper, the direct application of QDM and the combination of QDM with two occurrence-bias-adjusting methods, SSR and TDA, were compared with the focus on precipitation intensity, occurrence and the discharge simulated on the basis of the adjusted precipitation and evaporation. QDM is a robust method, but cannot adjust too dry simulations. In contrast with the direct application of QDM, both SSR and TDA are able to transform both too dry and too wet model simulations, necessary when considering continental-scale bias adjustment [35,36] or, when newer methods with an approach different from QDM are used, such as some multivariate bias-adjusting methods [38]. However, in the context of northwestern Europe, where simulations are generally too wet, it was shown theoretically that the assumptions allowing for the flexibility of both SSR and TDA could also result in a remaining bias in the number of dry days. SSR was initially designed to be applied in combination with CDF-t, whereas TDA has an important stochastic element. We explored this weakness in comparison with QDM, which can completely adjust an overabundance of wet days, but cannot be used with a preprocessing step when adjusting too dry simulations.

Although there were indeed small biases in the precipitation occurrence indices, these biases were much smaller than expected. On a total of 7305 days, only 17 days remained too wet after adjustment by SSR & QDM and only 43.7 days remained too dry after adjustment by TDA & QDM, where both numbers are an average based on 20 repetitions. Yet, the bias-adjusting methods did not perform similarly for all indices considered. A small bias remained for the precipitation lag-1 autocorrelation, whereas the wet-to-dry and dry-to-dry transition probabilities were well adjusted. This is probably caused by the internal structure of wet periods, which is only slightly affected by the occurrence-bias-adjustment. To attain a better structure of the wet periods, the application of convection-permitting models is promising [5,64,65]. Nonetheless, the remaining bias in lag-1 autocorrelation and in the number of dry days had almost no impact on the discharge simulation. Thus, both SSR and TDA perform better than expected and are well suited for hydrological impact assessment in the north-western European situation where the model is generally too wet.

Based on the good performance, the results in this study indicate that the more flexible methods are applicable in all situations and that they are very similar to each other. Note that these results are case-specific, and that care should be taken when they are extrapolated to other locations and especially other quantile mapping or even bias-adjusting methods. Nonetheless, we advise to use TDA rather than SSR. The latter is based on the CDF-t method, which is mathematically similar to the standard quantile mapping framework, but is based on different assumptions. Though the impact of using SSR was small in this study, it might be larger in other studies. The slightly larger bias in the number of dry days caused by TDA could be improved by tuning the parameter *b*, for instance on a monthly basis. However, the return of this considerable investment in the parameter is expected to be small. In any case, TDA is a promising alternative to SSR when adjusting occurrence biases at too wet grid boxes and should also perform properly for too dry grid boxes.

**Author Contributions:** Conceptualization, J.v.d.V., B.D.B. and N.E.C.V.; methodology, J.v.d.V., B.D.B. and N.E.C.V.; software, J.V.d.V.; validation, J.V.d.V., M.D., B.D.B. and N.E.C.V.; formal analysis, J.V.d.V.; investigation, J.V.d.V.; resources, J.V.d.V., M.D. and N.E.C.V.; data curation, J.v.d.V.; writing—original draft preparation, J.V.d.V.; writing—review and editing, J.V.d.V., M.D., B.D.B. and N.E.C.V.; visualization, J.V.d.V. and B.D.B.; supervision, M.D., B.D.B. and N.E.C.V.; project administration, N.E.C.V.; funding acquisition, B.D.B. and N.E.C.V. All authors have read and agreed to the published version of the manuscript.

**Funding:** This work was funded by FWO, grant number G.0039.18N.

**Institutional Review Board Statement:** Not applicable.

**Informed Consent Statement:** Not applicable.

**Data Availability Statement:** The code for the computations is publicly available [66]. The RCA4 data are downloaded and are available from the Earth System Grid Federation data repository. The local observations were obtained from RMI in Belgium, and cannot be shared with third parties.

**Acknowledgments:** The authors are grateful to the RMI for allowing the use of 117-year Uccle dataset. Jorn Van de Velde would like to thank Irina Y. Petrova for some comments regarding climate models. All authors would like to thank 2 anonymous reviewers for their helpful comments.

**Conflicts of Interest:** The authors declare no conflict of interest.

## Appendix A. Original Observed Values and Biases

**Table A1.** Observed values, and biases for the raw climate simulations, and the combinations of QDM and occurrence-bias-adjusting methods.

| | | Bias | | | |
|---|---|---|---|---|---|
| Index | Observed Value | Raw Climate Simulations | QDM | SSR & QDM | TDA & QDM |
| $Q_5$ (m$^3$/s) | 2.30 | 0.91 | $-0.33$ | $-0.33$ | $-0.32$ |
| $Q_{25}$ (m$^3$/s) | 3.36 | 1.44 | 0.01 | 0.01 | 0.01 |
| $Q_{50}$ (m$^3$/s) | 4.39 | 1.53 | 0.07 | 0.07 | 0.06 |
| $Q_{75}$ (m$^3$/s) | 5.72 | 2.50 | $-0.10$ | $-0.10$ | $-0.11$ |
| $Q_{90}$ (m$^3$/s) | 7.83 | 4.73 | $-0.39$ | $-0.39$ | $-0.41$ |
| $Q_{95}$(m$^3$/s) | 10.09 | 9.10 | $-1.05$ | $-1.05$ | $-1.06$ |
| $Q_{99}$ (m$^3$/s) | 18.71 | 18.24 | $-1.93$ | $-1.91$ | $-1.68$ |
| $Q_{99.5}$ (m$^3$/s) | 23.90 | 19.68 | $-0.71$ | 0.66 | 0.28 |
| $Q_{T20}$ (m$^3$/s) | 48.69 | 54.41 | 7.63 | 7.71 | 8.42 |
| $P_5$ (mm) | 0.00 | 0.00 | 0.00 | 0.00 | 0.00 |
| $P_{25}$ (mm) | 0.00 | 0.08 | 0.00 | 0.00 | 0.02 |
| $P_{50}$ (mm) | 0.10 | 1.01 | 0.05 | 0.05 | 0.04 |
| $P_{75}$(mm) | 2.70 | 1.83 | $-0.18$ | $-0.18$ | $-0.18$ |
| $P_{90}$(mm) | 7.40 | 1.99 | $-0.26$ | $-0.26$ | $-0.26$ |
| $P_{95}$ (mm) | 11.42 | 2.38 | $-0.61$ | $-0.61$ | $-0.61$ |
| $P_{99}$ (mm) | 21.80 | 2.38 | $-1.86$ | $-1.86$ | $-1.86$ |
| $P_{99.5}$ (mm) | 29.09 | 1.56 | $-4.20$ | $-4.20$ | $-4.20$ |
| $P_{P00}$ | 0.65 | 0.00 | 0.00 | 0.00 | $-0.01$ |
| $P_{P10}$ | 0.32 | 0.00 | 0.00 | $-0.00$ | 0.02 |
| $N_{dry}$ | 3470.00 | $-1466.00$ | 0.00 | $-17.00$ | 43.70 |
| $P_{lag1}$ | 0.33 | 0.11 | 0.03 | 0.03 | 0.03 |

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
