# Peer review of "Exploring the Effect of Occurrence-Bias-Adjustment Assumptions on Hydrological Impact Modeling"

_water, doi:10.3390/w13111573_

Round 1

Reviewer 1 Report

The review comments are attached.

Reviewer 2 Report

I think the topic is interesting and the paper is in general well written. My major concerns are the following:

-          The authors argue that Quantile Delta Mapping can adjust wet conditions but not dry conditions. They suggest SSR and TDA to resolve this problem. However the authors apply them in a case study with wet conditions. In fact the Results comparing the three approaches are similar (see Figures 3, 4, and 5). If the methodologies tested are indicated for dry conditions the authors should test them in these conditions. I can’t see the advantage of SSR and TDA for wet conditions.

-          At the end of the Introduction it is needed to include a paragraph to clarify the objectives and novelty of the research.

Others remarks:

Line 32: The spatial resolution of CORDEX RCMs is 10 km or around 12.5 Km?

Line 40-49: Include references about works where different approaches (delta change and bias correction) and techniques are used to correct RCMs. See:

  • Collados-Lara, A.J., Pulido-Velazquez, D., Pardo-Igúzquiza, E., 2018. An integrated statistical method to generate potential future climate scenarios to analyse droughts. Water (Switzerland). https://doi.org/10.3390/w10091224.
  • Pulido-Velazquez, D., Collados-Lara, A.J., Alcalá, F.J., 2018. Assessing impacts of future potential climate change scenarios on aquifer recharge in continental Spain. J. Hydrol. https://doi.org/10.1016/j.jhydrol.2017.10.077

Equation 1 an 2: Include explanation about the variables

Line 113-116: Only one RCM is used?

Figures 3, 4 and 5: I think it is difficult to see the information. I would increase the size.

Discussion and conclusions: I would present them in separate section.

Round 2

Reviewer 2 Report

Thank you for the responses to my previous comments. I think the manuscript is ready for publication.